# Design of Customized TPU Lattice Structures for Additive Manufacturing: Influence on the Functional Properties in Elastic Products

**DOI:** 10.3390/polym13244341

**Published:** 2021-12-11

**Authors:** Sergio de la Rosa, Pedro F. Mayuet, José Ramón Méndez Salgueiro, Lucía Rodríguez-Parada

**Affiliations:** 1Department of Mechanical Engineering and Industrial Design, Faculty Engineering, University of Cadiz, Puerto Real, 11519 Cádiz, Spain; pedro.mayuet@uca.es; 2Department of Civil Engineering, University College of Industrial Design, University of A Coruña, Ferrol, 15403 A Coruña, Spain; jrmendez@udc.es

**Keywords:** products design, elastic products, additive manufacturing, architect materials, lattice, TPU, material extrusion, customization

## Abstract

This work focuses on evaluating and establishing the relationship of the influence of geometrical and manufacturing parameters in stiffness of additively manufactured TPU lattice structures. The contribution of this work resides in the creation of a methodology that focuses on characterizing the behavior of elastic lattice structures. Likewise, resides in the possibility of using the statistical treatment of results as a guide to find favorable possibilities within the range of parameters studied and to predict the behavior of the structures. In order to characterize their behavior, different types of specimens were designed and tested by finite element simulation of a compression process using Computer Aided Engineering (CAE) tools. The tests showed that the stiffness depends on the topology of the cells of the lattice structure. For structures with different cell topologies, it has been possible to obtain an increase in the reaction force against compression from 24.7 N to 397 N for the same manufacturing conditions. It was shown that other parameters with a defined influence on the stiffness of the structure were the temperature and the unit size of the cells, all due to the development of fusion mechanisms and the variation in the volume of material used, respectively.

## 1. Introduction

Nowadays, the current market is characterized by continuous innovation and massive customization. This fact has increased the need for advanced techniques applied to the design of new products and the development of non-traditional manufacturing methods [1,2,3]. In this sense, Additive Manufacturing (AM) has been identified as a key point in manufacturing operations. AM has experienced extensive growth of almost 30% in the past 5 years [4], being considered one of the vital components of Industry 4.0 with sufficient potential to transform the global manufacturing industry [5,6,7].

AM technologies consisting of a set of manufacturing technologies based on the layer-by-layer controlled deposition of the material, directly from digital product data that contain information about the geometry. The layer-by-layer manufacturing method allows for more active interaction between the properties of the final product and the manufacturing parameters [8]. It also offers the ability to develop more complex geometries and structures with a high customization capacity, as well as a significant reduction in manufacturing time and cost [9]. These capabilities have made AM one of the most developed technologies to date, being very useful when low production volumes and frequent design changes are required [10,11,12], as well as the manufacture of early models and final products to bring them to market as soon as possible [4]. Currently AM is showing that it can contribute effectively to technological development in the future, being the focal point of research for industries such as footwear, automotive, medical and others [6,13,14,15]. Part of the development of this technology is also due to its compatibility with different types of materials, such as polymers, metals, ceramics, among others. From this group of materials, polymers have become the center of interest due to the versatility and wide range of properties they offer [16,17]. In particular, the existence of AM technologies compatible with elastomeric polymers such as the Material Extrusion Additive Manufacturing process [18] (also referred to as fused filament fabrication or FFF) [10,19], has caused an increase in their use. In addition, from the perspective of designing products with flexible properties, the ability of elastomeric materials to deform and regain their shape when a force has been applied (elasticity) is being of great interest for all types of applications in products and industrial sectors [20,21,22]. Among the different materials with elastic properties for FFF, a large number of them are based on Thermoplastic Polyurethane (TPU) [23,24,25]. The main reasons are their biocompatibility and their smoothness which allows the adhesion between layers in the impressions to be strong and durable [20,21,25].

The adaptability of the design and the high customization offered by AM open up new fields of application, increasing the interest in this material for the development of products that require a certain specific elasticity [16,17,26]. In sectors such as orthopedics and the medical field, the elasticity and the biocompatibility of TPU are especially useful for biomedical research [23,26,27,28]. Several lines of research focused on the characterization of this material already demonstrated the energy absorption capacity and compressibility of TPU for different filling conditions and thickness [28,29]. Tests with samples made of TPU with a filling density of 10% and a thickness of 5 mm in applications such as the case of a knee protector [28], showed that the product were suitable in terms of comfort, stability and flexibility, confirming that 3D printed TPU and similar materials could be used in personal in energy absorption applications [28].

Analyzing the current bibliography, it is found that the study of the functional properties related to the behavior conferred by the filling material and internal structure is especially relevant [30,31]. In this sense, FFF processes currently do not rival conventional manufacturing methods duo to the geometric freedom that allows them to make lightweight, high-performance components with the possibility of reducing weight, material consumption and modify the elastic properties of the material [4,32]. These internal structures offer the possibility of providing a different elasticity to the same product. Furthermore, they make it possible to obtain elastic materials from rigid materials and different elasticities from the same elastic material [26,33].

Under this pretext, in recent years, several geometric structures and solid models have been proposed and investigated in order to improve their effects and behavior based on their mechanical properties. The versatility and control over the manufacturing process offered by AM has made available a wide range of possibilities in the manufacture of lattice geometries and structures with more complex architectures [34,35,36,37]. This factor has led to a growing trend in the use of lattice structures, also due to the excellent energy absorption and impact properties that they can offer [30,38,39,40,41]. In addition this structures also demonstrated their potential for various applications from different industrial sectors [21,38], especially the possibility of adapting and optimizing materials for innovative lightweight and high-performance configurations [7,20].

Several lines of research focused on the energy absorption capacity of lattice structures and their use with elastic materials have been developed. It was demonstrated their ability to withstand repeated compressions until densification without failure. Similarly, it was shown that their energy absorption capacity depends on the deformation rate and the orientation of the cells with respect to the compression direction [41]. In the most recent works, the absorption of specific energy and the deformation behavior of extruded 2D lattices [42] and 3D lattice structures [36,43] have been evaluated, discovering that the variation of properties can effectively improve the specific energy absorption. An example of this research is the parametric study through experimentation and finite element of several lattice structures designs carried out by Fei Shen et al. [44]. In this research, different geometric parameters are used, showing that the resistance of this type of structure varies depending on the geometric parameters used. It is demonstrated that these parameters can exert influence on the variation of the density and porosity of the structure, increasing or decreasing the reaction force under the same maximum deformation [36,39]. In the same way, there are other lines of research focused on the characterization of gradual density structures. Is the case of the studies of graduated density versus normal density specimens carried out by I. Maskery et al. [45] and H. Niknam et al. [36]. In these studies, it was verified that the graduated structures exhibit a different compression behavior, being capable of absorb more energy per unit volume than their non-graded counterparts.

The variation according to the geometric parameters used (shape, density, geometry etc.) [46], manufacturing parameters, and variant of elastic material used [41], results in elastic behavior profiles that could be adapted to a specific application. This fact offer the possibility of designing elastic products with “customized” elastic properties for each range of energies in an efficient manner [39,45,46,47,48].This idea has been echoed in the branches of trauma medicine and rehabilitation. This is due to the fact that rehabilitation strategies are highly focused on a specific patient and must be carried out through specialized and personalized programs. These programs occasionally use specialized elastic products that offer specific resistance levels for each patient [49]. In this sense, the AM has provided unique advantages of flexibility and geometric freedom from a product design perspective. These advantages have allowed to improve adaptability and integrate functionality in order to solve discomfort problems and provide individualized support for obtain higher success rates [50].

According to the aforementioned, a high percentage of current studies are focused on the development of orthoses [51] and methodologies for the design of products for the localized treatment of different parts of the body [52,53]. In this sense, the rehabilitation takes place in static or resting conditions where the elastic properties of elastomeric materials are not used to the maximum of their potential. To a lesser extent, there are other studies that propose and study the use of the characteristics offered by elastomers and lattice structures in a more specific way. This is the case of products designed taking advantage of compression and energy absorption capacities. An example of this can be research focused on to the development of fully customized anatomical insoles for the foot [29,54,55,56]. In these studies, the light internal structures are used with the aim of modifying the shock absorption properties, providing the product with a greater degree of adaptation. to the particular damping needs of each user.

From the above approaches, the great potential offered by additive manufacturing in the development of customized elastic products can be deduced. However, the development of products with customized elastic properties is considered a field of little information, being potentially interesting for the development of more lines of research. Consequently, this study focuses on evaluating the influence of geometrical and manufacturing parameters on the stiffness properties of the material applied to lattice structures. Similarly, it focuses on establishing a relationship between the parameters and the properties offered. The aim is establishing a product design methodology that allows the medical needs of patients to be translated into parameters.

The relevance and contribution of this work lies in the creation of a working methodology that focuses on characterizing the behavior of an additively manufactured elastic lattice structures from geometrical and manufacturing parameters. In addition, the contribution resides in the possibility of using the statistical treatment results as a guide to find the most favorable possibilities within the range of parameters studied and to predict the behavior of the structures. These two previous statements mentioned are focused on taking advantage of the compressive and elastic characteristics to create specific functional properties in products. For all the above described, this work carried out the study of a total of 108 specimens, which were designed and tested under compression by finite element method (FEM) simulation. Likewise, a statistical treatment of the results of the tests was carried out, allowing to establish a relationship between the parameters and the behavioral properties offered by the specimens.

## 2. Experimental Procedure

The experiment was proposed with the aim of characterizing the influence of the geometrical and manufacturing parameters on the elastic behavior of lattice structures (Figure 1). Furthermore, this study also sought to characterize the specific properties for the design and development of customized products manufactured by AM.

### 2.1. Specimens

As mentioned before, the specimens were created to characterize their elastic properties according to the geometrical and manufacturing parameters of the FFF process. For this, several tests of resistance to compression were carried out. Different types of cell size cell topologies and manufacturing parameters were studied.

#### 2.1.1. Geometrical Parameters

The designed specimens were cubic, with an established fixed size of 9 cm × 9 cm × 9 cm. Each specimen was accompanied by an upper and lower bearing surfaces 2 mm thick (Figure 1).

A total of 4 different cell topologies have been studied. Two common geometric factors have been chosen from which different interactions and changes have been made (Figure 2). The exemplary topologies were modeled in Solidworks^®^ (Solidworks 2018, Dassault Systèmes, Velizy-Villacoublay, France) computer-aided design software:BCC—Body Centered CubicBCCz—Body Centered Cubic with columns in the z directionFCC—Face Centered CubicFBCC—Face and Body Centered Cubic

Three different unit cell sizes have been studied: 1.5 cm, 3 cm, 4.5 cm. All the unit cell sizes are equal in x-y-z directions, where x-y are directions in the plane, and y-z is the thickness direction. Each unit cell is a cube of 1.5 mm × 1.5 mm × 1.5 mm, 3 mm × 3 mm × 3 mm, 4.5 mm × 4.5 mm × 4.5 mm, respectively (Table 1). According to the lattice unit size, the number of unit cells is repeated to maintain the original dimensions of the structure of 9 cm × 9 cm × 9 cm. This mean that for a size of 1.5 cm the structure is 6 × 6 × 6 unit cells, for 3 cm it is 3 × 3 × 3 unit cells, and for 4.5 it is 2 × 2 × 2 unit cells (Table 1). In each of the unit sizes the thickness of the walls is kept constant at a size of 1.20 mm thick.

#### 2.1.2. Manufacturing Parameters

Similarly to [57,58], different raster orientation angles (R) have been investigated. In particular, three R have been chosen: 20°, 45° and 90°. For each R, three different extrusion temperatures (T) have been set: 180 °C, 190 °C and 200 °C for the TPU. All temperatures adopted are above the material melting point.

### 2.2. Materials

The material used in this study was the commercial TPU 95A filament from the Ultimaker company (Utrecht, The Netherlands) [59]. The mechanical properties given by the manufacturer are detailed in Table 2.

From [57,58], it is possible to observe that the behavior of the material against the variation of the T and R parameters is similar in both cases. From the results of [57,58], firstly, the percentage of variation of E (Young’s modulus) was extracted for each value of T and R by using Equation (1).
(1)$$E=\frac{d\sigma }{d\varepsilon }$$

Secondly, the percentage of variation of *E* with respect to the *E* given by the manufacturer in both articles was compared, finding that the variation is similar. Taking advantage of this fact, it was possible to extract an approximate model of the behavior of the TPU material.

Lastly, applying the variation percentages to the material chosen for this study, the most representative values have been obtained for the creation of the materials (Young’s Modulus (*E*) and elastic limit (*σy*)). Subsequently, the percentages have been applied to the digital models of the specimens during the virtual tests (Table 3).

### 2.3. Procedure of Evaluation with CAE Simulation

A total of 108 tests were carried out. The number of iterations is shown in Table 4.

The nomenclature of each test is represented according to the following formulation:G(*)_R(*)_T(*)_U(*)

G corresponds to the type of cell geometry.R corresponds to the orientation angle of the fill.T corresponds to the temperature.U corresponds to the unit size.(*) Value of the associated parameter

As an example, the nomenclature for testing a BCC geometry specimen whose U = 3 cm, R = 90° and T = 190 °C would be GBCC_R90_T190_U3.

As previously mentioned, the simulations of the compression tests were carried out using the Finite Element method (FEM). The SolidWorks^®^ (Solidworks 2018, Dassault Systèmes, Velizy-Villacoublay, France) software was used, both for the modeling of the samples and for the simulation.

First, the specimen material was defined. For this, custom elastic materials were created by entering the properties according to Table 3.

Second, the test boundary conditions were defined. The type of test that was performed was a non-linear static compression test. It has been considered that each specimen is fixed to the soil surface by one of its bearing surfaces. In addition, a displacement of 1/3 of the initial sample size has been imposed on the opposite bearing surface in the perpendicular direction (Figure 3).

The parameter chosen to discuss the results has been the Reaction Force (from now RF). This term can be defined as the equal and opposite force exerted by the specimen in response to compression deformation. The objective has been to determine what force must be applied to each specimen to compress 1/3 of its initial size. Depending on the force required, specimens of greater or lesser stiffness have been distinguished.

An evaluation of the data obtained has been carried out by a statistical treatment. For this, the Minitab (Minitab 2018, Minitab LLC, State College, PE, USA) software has been used. This is a software designed to work with basic and advanced statistical functions. The statistical treatment carried out has been composed of the following elements:Bar graph of the global results to visualize the variation of RF values.Interaction graphs to analyze the relationship between the variables and their influence on the specimens.ANOVA (Analysis of variance) to evaluate the importance of each parameters.Main effects graph to observe the way in which each variable affects the behavior of the specimens.Contour plots to evaluate the most favorable possibilities within the range of parameters studied.

## 3. Results and Discussion

The results of each specimen are shown in the diagram of Figure 4. This figure has been extracted from Appendix A, specifically from Table A1, Table A2, Table A3 and Table A4 where a more detailed view of the results grouped by geometry type is displayed.

In a first impression of the results, it is possible to observe that the values of RF are highly variable and clearly depend on the cell topology. BCC type is the one that offers the smallest values, being BCCz the one that offers the highest values, followed by FBCC and FCC. The range of minimum and maximum values offered by the BCC type is [2365 N–33.8 N] while for the other types of geometry are [40,284 N–543 N] for BCCz, [13,624 N–282 N] for FCC, and [16.8 N–369 N] for FBCC. This means that the maximum and minimum values obtained from BCCz increase approximately 1500% respect to BCC, and approximately 97% and 51% respect to FCC and FBCC, respectively. This increase in values may be related to the geometric complexity of each type of cell topology and its compression capacity. Accordingly, the morphological analysis will discuss these differences in greater detail.

Another factor to highlight is that for all types of cell topology studied, it is observed that the tests whose combination of parameters is R = 90°, T = 200 °C and U = 1.5 cm offer unfavorable results. Unfavorable tests have been considered to those whose simulation predicts that the structures do not support compression stresses (Those tests where the sample presents tensions higher than the yield strength of the material). For a more detailed explanation of the term unfavorable test, see Figure A1 from Appendix A.

These unfavorable tests may be due to the fact that the combination of these parameters can result in a much stiffer structure that does not withstand compression forces, causing the specimens to break. Accordingly, these differences will be discussed in more detail in the elasticity analysis.

### 3.1. Morphological Analysis

Emphasizing the behavior under compression of each type of cell topology (Figure 5), a relationship between the compression capacity, the accumulation of stresses, and the geometric complexity of the cell (number of junction nodes and beams) is observed.

BCC type is the one with the least accumulated stresses. This may be due to the fact that it is geometrically simpler than the rest of the cell topologies and it only has one junction node. This would result in a greater deformation capacity, requiring a lower energy input for its compression, resulting in lower RF values. In the opposite part, FBCC presents a greater accumulation of stresses. This may be due to the fact that FBCC has the highest geometric complexity (Highest number of junction nodes and beams).

As is mentioned in the compressive tests of different additively manufactured lattice morphologies carried performed on [60], the number of horizontal, vertical and inclined beams in the structures and their thickness, has influence over the compression behavior and the buckling of the structures. This would explain why FBCC presents a greater accumulation of stresses (Figure 5c) compared to BCC (Figure 5a) and FCC (Figure 5b). It is possible to conclude that, the number of nodes and beams of FBCC hinders the deformation capacity of the cell topology. This causes the cell to exert a greater opposition to the compression forces (greater reaction force) being necessary to apply a higher quantity of energy for its compression. This would explain why FBCC presents higher RF values than FCC, and why BCC presents such low values with respect to the rest of cell topologies. However, as seen previously, an important point to note is that although FBCC has the highest geometric complexity, BCCz (Figure 5d) presents the highest RF values.

Analyzing the behavior of BCCz cell type separately, it is observed that the accumulation of stresses appears in a much more concentrated way compared to the rest of the geometry, especially in the area of vertical geometric elements. This behavior may be due to the fact that the columns are vertical structural elements whose compression direction coincides with the direction of the applied compression forces. As mentioned in [61], vertical members potentially influence stiffness and the amount of strain energy absorbed during compression. This would conclude why it is necessary to apply a greater energy for its compression (Figure 5d) and why it presents the highest RF values.

### 3.2. Elasticity Analysis

Analyzing in more detail the influence of the other parameters on the compression behavior of the specimens, a statistical treatment of the data has been carried out. Several interaction graphs have been generated to identify the growth and decrease of trends between the variables studied. The interaction graphs have been made for each type of cell topology, and they are all represented in Appendix B, specifically in Figure A2, Figure A3, Figure A4 and Figure A5.

Analyzing the interaction graphs, it is observed that the results are very similar. Specifically, the way in which each parameter is displayed is similar, but each of these relationships is adapted to the range of RF values offered by each cell topology (Figure A2, Figure A3, Figure A4 and Figure A5)

The similarity between the graphs may be derived from the nature of the simulation tests. In simulated compression, variables of a physical nature from non-virtual testing which may influence the behavior of the specimen and the plots graphs, are totally absent. That is why only the graphs corresponding to the FCC type cell has been taken as a reference for the analysis.

The interaction of the Unit size (U) parameter is represented in Figure 6 (extracted from the interaction graph of Figure A4 in Appendix B). It is observed that U exerts a great influence on the RF value. In the same way, RF value tends to decrease as U increases. One factor to highlight is that the RF value decreases much more notably when the U varies from 1.5 cm to 3 cm (decreases by 80%) compared to when U varies from 3 cm to 4.5 cm (decreases by 47%). This decrease in the RF value can be justified by the porosity of the structure. For higher values of U, the specimens are much more porous and have less material volume with respect to the total volume of the structure (Table 1). Similarly, when the value of U is higher, the number of cells that are repeated to form the structure, is reduced by 50% for a value of U = 3 cm, and 30% for U = 4.5 cm (With respect to U = 3 cm). This would explain why the value of RF varies much more notably when U varies from 1.5 cm to 3 cm and vice versa.

It could be concluded that, for smaller U values, the porosity of the structure decreases and the number of cell units would increase (and vice versa). This provides greater structural strength to the specimen which translates into greater resistance to deformation, being necessary to apply more energy to compression and resulting in higher RF values.

The interaction of the Temperature (T) parameter is represented In Figure 7 (extracted from the interaction graph of Figure A4 in Appendix B). It is observed that the highest RF values have been obtained for T = 200 °C. RF values for T = 180 °C and T = 190 °C are more distant from the values obtained for T = 200 °C. Note that the change in value from T = 180 °C to T = 190 °C causes a decrease in RF values by 15%, while from T = 190 °C to T = 200 °C increase by 55%. The highest values obtained for T = 200 °C may be due to a better melting of the filling material and consequently a better adhesion of the layers. This would notably increase the mechanical resistance of the structure as is exposed in [57,58], being necessary to apply more energy to compress the specimen and resulting in higher RF values. Likewise, the fact that the values for T = 180 °C and T = 190 °C are lower than those of T = 200 °C, may be due to the absence of the melting mechanism for those temperatures.

It would be logical to conclude that the values of RF increase linearly as T does. However, the values of RF for T = 190 °C are less than those for T = 180 °C. This fact can be especially related to the parameter R. Analyzing the difference between the values of RF of T = 190 °C with respect to those of T = 180 °C, that difference becomes much more noticeable when T is related to R, especially for R = 45° (Figure 7b). With this pretext, the influence of this parameter on the structure has been analyzed in greater detail.

The interaction of the Raster Angle Orientation (R) parameter is represented In Figure 8 (extracted from the interaction graph of Figure A4). According to the interaction graph, the highest RF values have been obtained for the orientation R = 20°, followed by R = 90° and R = 45°. According to [58], the mechanical resistance of the specimens according to the different angles, can be influenced by the mechanism of fusion of the filling material. Similarly the structural behavior can be reduced by the formation of inclusions and internal defects such as bubbles within the the layers [58]. It is notable that the values of RF decrease for R = 45°. This may be due to the fact that, for this orientation the production of inclusions can develop more predominantly than for other orientations [58]. This fact would negatively affect the mechanical behavior. Referring to the aforementioned, this mechanism could be the reason why the values obtained for T = 190 °C were lower than those obtained for T = 180 °C.

It is important to highlight that the highest values have been obtained for orientations close to R = 0° and R = 90°, whose values were similarly close. According to [58], for orientations close to R = 0° both mechanisms (melting and inclusion of defects) exert influence over the behavior of the specimens. In the same way, for the orientation R = 90° melting mechanism plays a more relevant role with respect to the presence of defects. In Figure 8, the RF values obtained for R = 20° and R = 90° are similar, being higher for R = 20°. However, in Figure 8b, it is possible to observe that for T = 200 °C the value of RF for R = 90°, exceeds the value for R = 20°. As mentioned before, this may be due to the fact that for R = 90 °C the melting of the filling material appears more frequently, as in the case for T = 200 °C. Therefore, it is possible that the combination of both parameters results in a more homogeneous and stiffer structure. This could explain the presence of failed simulations where the structures did not support the compression loads.

When the parameters T = 200 °C and R = 90° relative to each other, and the parameter U = 1.5 cm is also introduced, then the theoretically stiffest structure of topology cell would be obtained. This would be enough to fail during compression.

It is possible to conclude that the RF values will always be higher for the orientations of R = 20° and R = 90° where the melting mechanism can play an important role respect to the presence of defects. On the other hand, it is concluded that the R = 45° values will be lower, possibly due to the development of inclusions in the fibers of the structure that seem to reduce their resistance to compression.

The influence of each parameter on the structure is also corroborated by the ANOVA analysis represented below in Table 5.

Analyzing F-value, the data show that the most important parameters are U and T, with a mean value of F = 103.04 and F = 9.88, respectively. This mean as expected, that the parameter U seems to have a much more defined influence on the value of RF, especially when U decreases from 3.0 cm to 1.5 cm. This is reflected in the slope from the main effects graph in Figure 9 (Obtained from BCC cell topology and similar for the rest of the topologies). In the same way, another increase in the slope was also detected from T = 190 °C to T = 200 °C. The parameter R seems to have a less defined influence with respect to the rest of the parameters.

Analyzing *p*-value, the data show a mean value for U and T of *p* = 0.000 and *p* = 0.002, respectively, being higher for R with a *p*-value of and *p* = 0.151. This mean, U and T results are more reliable, indicating the probability of obtaining a similar value in a similar test is higher. Consequently, this mean R results are more arbitrary with a lower probability of obtaining similar results in similar tests. This could explain why R seems to have a less defined influence over the RF value with respect to the rest of the parameters.

Finally, contour plots have been made to establish response values and desirable operating conditions. Each of the contour plots for each topology cell type are represented in Appendix C, especially in Figure A6, Figure A7, Figure A8 and Figure A9. The contour plots for all cell topologies show a similar behavior and corroborate the conclusions previously obtained. Overall, it is observed that the maximum RF values are obtained when U decreases and T approaches 200 °C, as well as when R approaches 20 °C (Figure 10).

As revealed in the ANOVA, it is possible to visualize that the most significant variable is U. In the same way it is observed that the influence of the rest of the parameters increases when U decreases, and vice versa. An example of this can be seen in Figure 10a,b (both extracted from the interaction graph of Figure A8), which reveals that from U = 3.5 cm T and R have practically zero influence on the value of RF.

The possibility of achieving the same reaction force using different parameters has been detected. For example, it has been found that for all cell topologies the GX_R20_T190_1.5 and GX_R90_T180_1.5 tests (understand X as any of the 4 topology cell types studied) present practically identical results.

This confirms that contour graphs can be a very useful and interesting guiding tool to find different existing possibilities of reaching the same value of RF. Furthermore, to extract or predict the possible behaviors of the structures other unit sizes, temperatures and raster orientation angles. In addition, the choice of one way or another can have different advantages in printing time and material waste during the specimens manufacturing stage.

## 4. Conclusions

In this work, the evaluation of the influence of the main geometric and manufacturing parameters on the properties of additively manufactured TPU lattice structures is presented. This work explores a field not studied so far which focuses on the design of flexible custom products using lattice structures. The capabilities to adapt to design changes and create elements with variable repetitive geometries offer an opportunity to develop flexible elements that can customize the degree of stiffness of certain areas of the same element. Therefore, this line of research is key for the characterization of elastic parameters according to the structure of the product. The procedure presented here for the custom design of elastic products offers great advantages in the field of new product design. Thanks to the design of lattice structures specifically conceived for the designed product, a new line of work is opened and can be applied to different fields such as consumer or healthcare products.

According to the results, in terms of geometric parameters, it is concluded that the RF values of the specimens are highly variable and clearly depend on the cell topology, influencing the variation of the final elasticity of product. The stress accumulations of each type of cell topology depend on its geometric complexity. It was detected that a greater number of geometric elements and node points hinders the deformation capacity of the geometry. This causes the structure to exert greater opposition to the compression forces causing an increase in the values of RF. Similarly, it was observed that the presence of vertical structural components have a significant influence on the increase in RF values. This is because the direction of the columns coincides with the direction of the compressive stresses in the simulation, increasing the stress concentration and exerting greater opposition to the compressive forces.

Regarding the rest of the parameters, it is concluded that U have a much more defined influence on the value of RF with respect to the parameters T and R. The influence of U can be justified by two aspects; Firstly, for higher values of U the samples are more porous and have less material volume with respect to the total volume of the structure. Secondly, the number of lattice units that are repeated to make up the structure is less. Both aspects have a direct influence on the density of the structure determining that the variation in density is the factor with the greatest influence on the stiffness of the structure. This is corroborated by the contour graphs, where it is represented that the influence of the rest of the parameters increases when U decreases, and vice versa. Therefore, from a product design perspective, it could be said that the force required to compress the geometry is highly dependent on the density of the structure and the geometry complexity of each cell topology.

Regarding the parameter T and R, it is concluded that the highest RF values are obtained for higher temperatures such as T = 200 °C. This is due to the appearance of mechanisms of melting of the filling material. This fact improves the adhesion of the layers and notably increasing the mechanical resistance of the specimens. Regarding R, it is concluded that it has a less defined influence with respect to the rest of the parameters. The values of RF decrease notably for R = 45°. This may be due to the appearance of inclusions or air bubbles much more predominantly for this orientation. For R = 20° both mechanisms intervene (Fusion of layers and inclusion of defects) and for R = 90° the fusion of fibers plays a more relevant role with respect to the presence of defects. The tests whose results have not been satisfactory may be due to the combination of these parameters used. By using the parameter U = 1.5 cm in combination with the parameters T = 200 °C and R = 90°, the theoretically stiffest structure of each cell topology would be obtained. The RF values would be sufficiently high to cause the failure of the specimen during compression.

For another part, it is concluded that contour graphs can be a very useful guide tool to find different existing possibilities of reaching the same value of RF and to predict the behavior of the structures from parameters. In addition, they can serve as an orientation to use new parameters not studied for reaching new values of RF. The choice of a specific group of parameters can bring different advantages in printing time and waste of material during the manufacturing stage of specimens with the same stiffness.

Due to the fact that the present study is a theoretical study with a great content of virtual simulation, the rheological parameters have not been taken into account. It is known that the rheological characteristics of the viscous polymer when it cools once deposited on the printing bed, govern the degree of interlayer welding impacting the mechanical performance of the printed parts. Controlling and monitoring rheological properties, such as zero shear viscosities and melt shear moduli, is of great importance in this region to ensure adequate mechanical robustness and shape integrity of the deposited layers.

As a future line of research, it is considered of interest to deepen the monitoring of rheological properties and deepen the analysis of internal structures and densities adapted to the specific purpose of the product. Likewise, it would be interesting to consider the possibility of implementing different types of geometries in the same product.

Due to the existing customization needs in sectors such as trauma medicine and rehabilitation, a field of research remains open to work on the properties of additively manufactured elastic lattice structures. The final objective of the research would be oriented to the development of fully functional custom elastic products and to increase the performance of additive manufacturing with TPU and the final quality of these products.

## Figures and Tables

**Figure 1 polymers-13-04341-f001:**
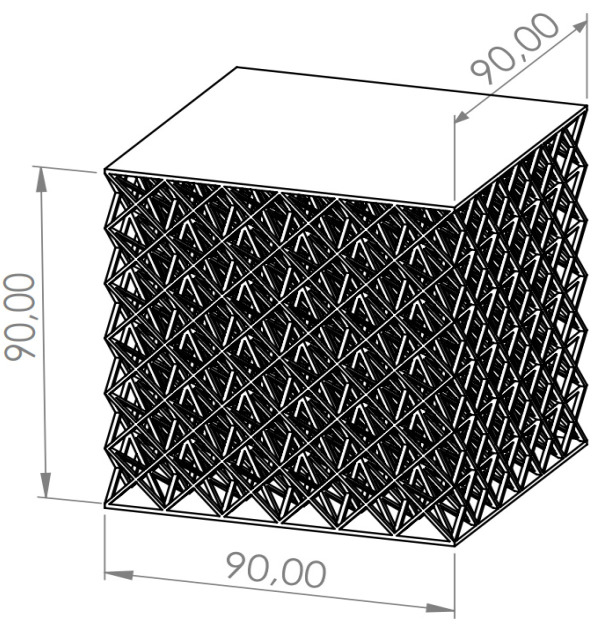
Lattice structure dimensions (mm).

**Figure 2 polymers-13-04341-f002:**
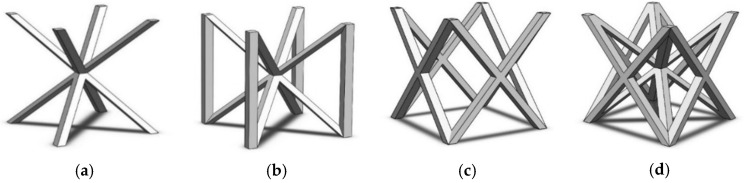
Cell Topologies. (**a**) BCC; (**b**) BCCz; (**c**) FCC; (**d**) FBCC.

**Figure 3 polymers-13-04341-f003:**
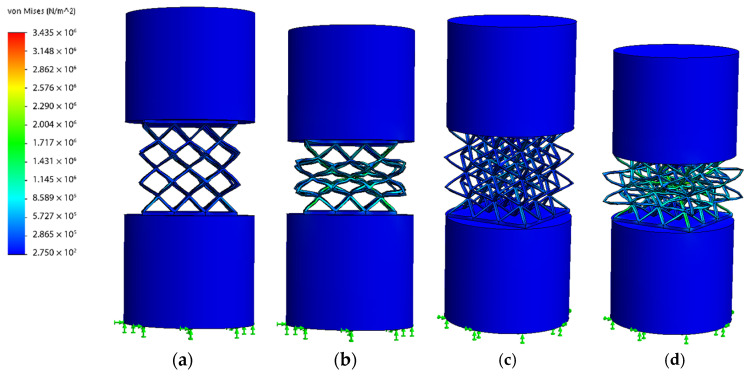
Simulation of the specimen in compression; (**a**) Inicial state Frontal view: (**b**) Specimen in compression state (Frontal view); (**c**) Inicial state (Perspective view); (**d**) Specimen in compression state (Perspective view).

**Figure 4 polymers-13-04341-f004:**
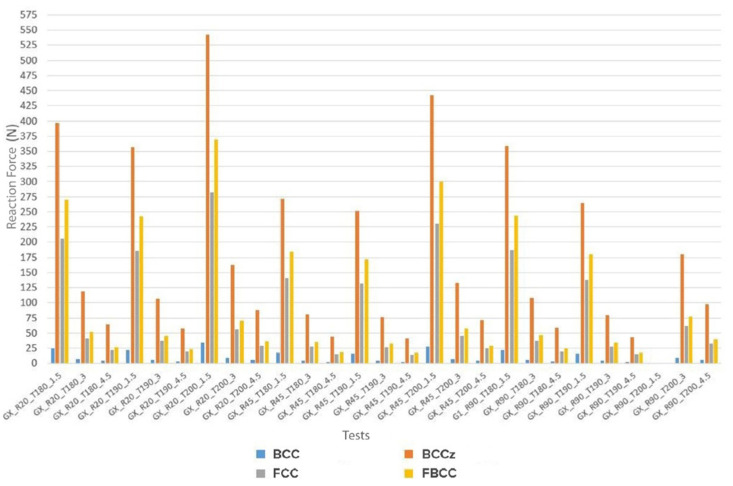
Reaction forces of the different cell topologies. In the term “GX”, X refers to the type of cell topology.

**Figure 5 polymers-13-04341-f005:**
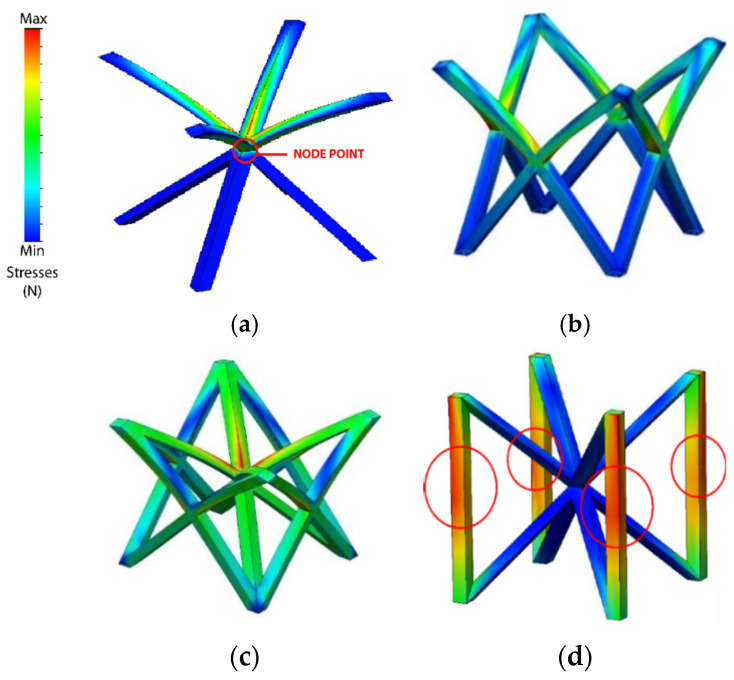
Compression stress analysis of the different cell topologies; (**a**) BCC; (**b**) FCC; (**c**) FBCC; (**d**) BCCz.

**Figure 6 polymers-13-04341-f006:**
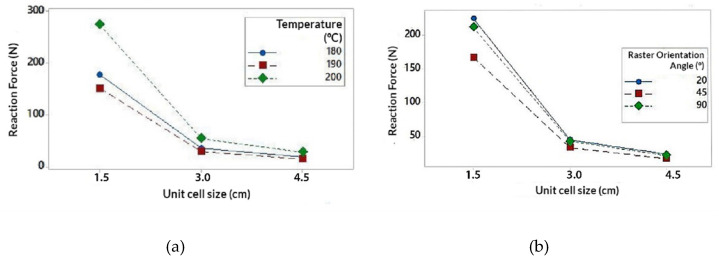
Interaction graph on unit size (U); (**a**) Unit size (U) in relation to temperature (T); (**b**) Unit size (T) in relation to the orientation angle (R).

**Figure 7 polymers-13-04341-f007:**
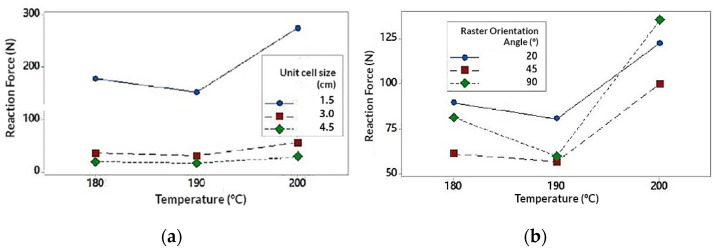
Interaction graph on temperature (T); (**a**) Temperature (T) in relation to unit size (U); (**b**) Temperature (T) in relation to the orientation angle (R).

**Figure 8 polymers-13-04341-f008:**
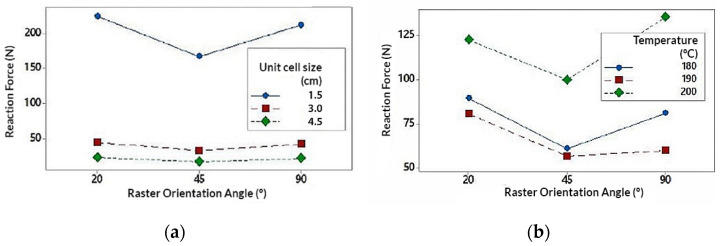
Interaction graph on unit size (U); (**a**) Orientation angle (R) in relation to unit size (U); (**b**) Orientation angle (R) in relation to temperature (T).

**Figure 9 polymers-13-04341-f009:**
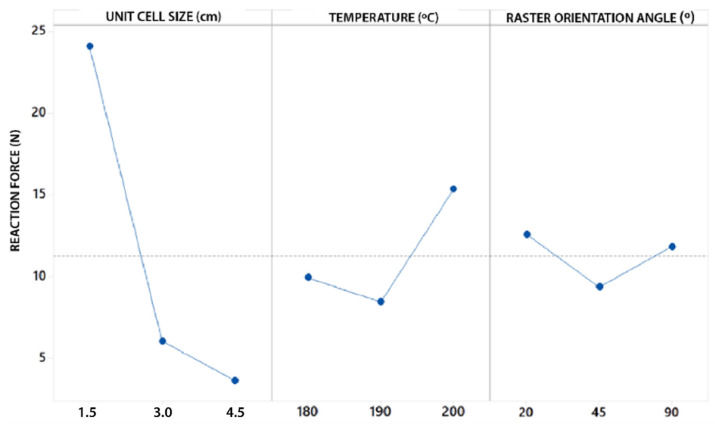
Main effects graph.

**Figure 10 polymers-13-04341-f010:**
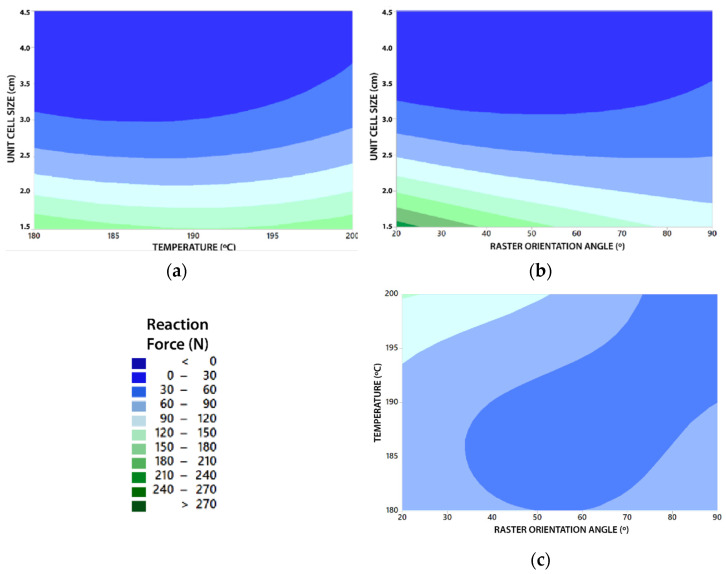
Contour plot (Example of FCC cell type study); (**a**) Unit size (U) vs. Temperature (T); (**b**) Unit size (U) vs. Fill orientation (R); (**c**) Temperature (T) vs. Fill Orientation (R).

**Table 1 polymers-13-04341-t001:** Unit cell size dimensions (cm)—FCC structure example.

Unit Cell Size 1.5 cm	Unit Cell Size 3 cm	Unit Cell Size 4.5 cm
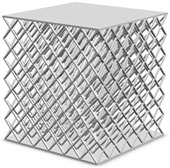	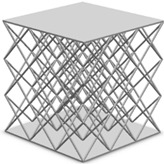	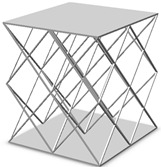
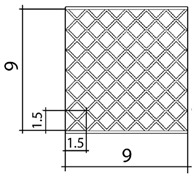	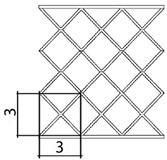	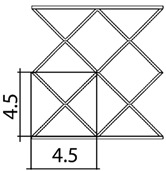

**Table 2 polymers-13-04341-t002:** Main mechanical and physical properties of TPU, provided by Manufacturer [59].

Material	Density (g/cm^3^)	Tensile Strength (MPa)	Melting Temperature (°C)	Elasticity (MPa)	Shore Hardness A
TPU	1.02	39	220	26	95

**Table 3 polymers-13-04341-t003:** TPU properties according to the Temperature and Orientation Angle.

	180 °C		190 °C		200 °C	
	*E* (Mpa)	*σy* (Mpa)	*E* (Mpa)	*σy* (Mpa)	*E* (Mpa)	*σy* (Mpa)
20°	13.9	8.5	12.5	8.45	19.6	10.6
45°	9.5	5.6	8.8	5.6	15.5	7.7
90°	12.6	6.8	9.3	5.8	21	8.2

Density 1020 kg/m^3^ (Provided by Manufacturer).

**Table 4 polymers-13-04341-t004:** Number of tests.

Parameters	Values
Cell Topology (G)	BCC–BCCz–FCC–FBCC
Unit cell size (U)	1.5 cm–3 cm–4.5 cm
Raster orientation angle (R)	0°–45°–90°
Printing Temperature (T)	180 °C–190 °C–200 °C
Total	108 test

**Table 5 polymers-13-04341-t005:** ANOVA (Analysis of variance) of each parameter.

U (cm)			T (°C)			R (°)		
Variable	F-Value	*p*-Value	Variable	F-Value	*p*-Value	Variable	F-Value	*p*-Value
RF(N)-BCC	101.1	0.000	RF(N)-BCC	10.74	0.001	RF(N)-BCC	2.29	0.127
RF(N)-BCCz	100.82	0.000	RF(N)-BCCz	12.28	0.000	RF(N)-BCCz	2.62	0.097
RF(N)-FCC	105.02	0.000	RF(N)-FCC	8.44	0.002	RF(N)-FCC	1.84	0.185
RF(N)-FBCC	104.44	0.000	RF(N)-FBCC	8.08	0.003	RF(N)-FBCC	1.76	0.198

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
