# Peer review of "Design of Customized TPU Lattice Structures for Additive Manufacturing: Influence on the Functional Properties in Elastic Products"

_polymers, 2021, doi:10.3390/polym13244341_

Round 1

Reviewer 1 Report

  • The authors propose an investigation method for industrially importance in the lattice materials research . Both method and the material are of high scientific interest.
  • Add to the abstract quantified results. It says greater stiffness, but how much more? In compared to what? It is very vaguely written.
  • In the introduction the authors explain very well the importance of the process and the material for additive manufacturing industry. They provide a very complete state of the art with a lot of qualitative data. I recommend including more quantitative results of other authors. This will help to make their results stronger. The introduction is very complete with minor details.
  • In the experimental part, I recommend developing more the methodology or equations that they used to get the approximate behavior law of the TPU. Because is not clear if it is a model, prediction, or empirical determination. Also, in the table they stablish the value of the density both they did not mention if this value is part of the approximation or is part of the technical data provided by the manufacturer of the TPU.
  • Section 2.3 is named “Procedure of evaluation with CAE simulation” but the authors never establish this abbreviation before. Even is very commonly known is important to always establish your abbreviations the first time you use them.
  • Also in section 2.3, in the line 228 the authors write “temperature” instead of temperature. Is important to review all the language in the manuscript to avoid this type of errors.
  • Line 236 the name of the software “Solidworks” is misspelled
  • In line 282 when the authors mentioned the “unfavorable results” I recommend including an example to make this statement stronger.
  • For the morphological analysis, I recommend including references of other authors to make the statements stronger.
  • The elasticity analysis is very well explained with minor corrections. Check to spelling and grammar in this part.
  • For a very large experimental design with so many samples, there are other statistical methods stronger that ANOVA. If you apply one of those you can have a stronger statistical model.
  • As using TPU the rheological parameters of the material will play a very important role in the FFF process and the elasticity behavior of the lattice structures. The authors don’t even mention these variables. If the rheology is not taken on account is important to mention, why they decided to not take it into account.

Author Response

The authors propose an investigation method for industrially importance in the lattice materials research. Both method and the material are of high scientific interest.

Add to the abstract quantified results. It says greater stiffness, but how much more? In compared to what? It is very vaguely written.

ANSWER: Done.

In the introduction the authors explain very well the importance of the process and the material for additive manufacturing industry. They provide a very complete state of the art with a lot of qualitative data. I recommend including more quantitative results of other authors. This will help to make their results stronger. The introduction is very complete with minor details.

ANSWER: More quantitative results from other authors have been introduced in the introduction.

In the experimental part, I recommend developing more the methodology or equations that they used to get the approximate behavior law of the TPU. Because is not clear if it is a model, prediction, or empirical determination. Also, in the table they stablish the value of the density both they did not mention if this value is part of the approximation or is part of the technical data provided by the manufacturer of the TPU.

ANSWER: The explanation of the methodology carried out to obtain the approximate behavior model of the TPU has been improved. It has been indicated which are the technical data provided by the manufacturer,

Section 2.3 is named “Procedure of evaluation with CAE simulation” but the authors never establish this abbreviation before. Even is very commonly known is important to always establish your abbreviations the first time you use them.

ANSWER: It has been mentioned again earlier in the text, and the explanation of the acronym has been added.

Also in section 2.3, in the line 228 the authors write “temperature” instead of temperature. Is important to review all the language in the manuscript to avoid this type of errors.

ANSWER: Done.

Line 236 the name of the software “Solidworks” is misspelled

ANSWER: Done.

In line 282 when the authors mentioned the “unfavorable results” I recommend including an example to make this statement stronger.

ANSWER: A more detailed explanation has been included in Appendix A and referenced in the results and discussion.

For the morphological analysis, I recommend including references of other authors to make the statements stronger.

ANSWER: Done.

The elasticity analysis is very well explained with minor corrections. Check to spelling and grammar in this part.

ANSWER: The text has been revised and corrected

For a very large experimental design with so many samples, there are other statistical methods stronger that ANOVA. If you apply one of those you can have a stronger statistical model.

ANSWER: The meaning of the p-value and its relation to the results obtained has been improved and discussed. With this explanation, it is considered that the statistical model is better defined.

As using TPU the rheological parameters of the material will play a very important role in the FFF process and the elasticity behavior of the lattice structures. The authors don’t even mention these variables. If the rheology is not taken on account is important to mention, why they decided to not take it into account.

ANSWER: A comment about the rheological parameters of the material and why those parameters have not been taken into account, has been added.

Reviewer 2 Report

The manuscript “polymers-1476119-v1” dealing with 3dprinting of FFF (MEX) has been refereed. Please comments are listed below:

  1. Use ASTM 52900 for correct terminologies of AM processes. FFF is the commercial name of Stratasys and the correct terminology is Material Extrusion (MEX).
  2. Suggest removing figure 1 from the introduction.
  3. Additive manufacturing now has many advantages over conventional manufacturing. To highlight your work, add a short note in the introduction by using the following paper and mention the privilege of additive manufacturing. “Additive manufacturing a powerful tool for the aerospace industry” 2021.
  4. Highlight the contribution of the paper and make it bolder.
  5. Add P-Values clearly and discuss that.
  6. The explanation in Figure 4 is not sufficient enough. Please add more mechanisms for this figure and the next figure.
  7. Update the introduction by comparing the following new references on PLA printing with reviewed paper in your article.

  • Kumar Mishra, P., S. Ponnusamy, and M.S. Reddy Nallamilli, The influence of process parameters on the impact resistance of 3D printed PLA specimens under water-absorption and heat-treated conditions.
  • von Windheim, N., D.W. Collinson, T. Lau, L.C. Brinson, and K. Gall, The influence of porosity, crystallinity and interlayer adhesion on the tensile strength of 3D printed polylactic acid (PLA)
  • Afonso, J.A., J.L. Alves, G. Caldas, B.P. Gouveia, L. Santana, and J. Belinha, Influence of 3D printing process parameters on the mechanical properties and mass of PLA parts and predictive models
  • Gonzalez Alvarez, A., P.L. Evans, L. Dovgalski, and I. Goldsmith, Design, additive manufacture and clinical application of a patient-specific titanium implant to anatomically reconstruct a large chest wall defect.
  • Travieso-Rodriguez, J.A., R. Jerez-Mesa, J. Llumà, G. Gomez-Gras, and O. Casadesus, Comparative study of the flexural properties of ABS, PLA and a PLA–wood composite manufactured through fused filament fabrication.

Author Response

  1. Use ASTM 52900 for correct terminologies of AM processes. FFF is the commercial name of Stratasys and the correct terminology is Material Extrusion (MEX).

ANSWER: The terminology was mentioned and corrected.

  1. Suggest removing figure 1 from the introduction.

ANSWER: It has been removed from the introduction and introduced in the experimental procedure.

  1. Additive manufacturing now has many advantages over conventional manufacturing. To highlight your work, add a short note in the introduction by using the following paper and mention the privilege of additive manufacturing. “Additive manufacturing a powerful tool for the aerospace industry” 2021.

ANSWER: The revision of the article has been added in the introduction.

  1. Highlight the contribution of the paper and make it bolder.

ANSWER: The contribution has been highlighted and made bolder in both the abstract and the introduction.

  1. Add P-Values clearly and discuss that.

ANSWER: P-Values were added clearly and their relationship with the results obtained has been discussed.

  1. The explanation in Figure 4 is not sufficient enough. Please add more mechanisms for this figure and the next figure.

ANSWER: The Figure has been modified, and the explanation of the figure legend has been extended.

  1. Update the introduction by comparing the following new references on PLA printing with reviewed paper in your article.

Kumar Mishra, P., S. Ponnusamy, and M.S. Reddy Nallamilli, The influence of process parameters on the impact resistance of 3D printed PLA specimens under water-absorption and heat-treated conditions.

von Windheim, N., D.W. Collinson, T. Lau, L.C. Brinson, and K. Gall, The influence of porosity, crystallinity and interlayer adhesion on the tensile strength of 3D printed polylactic acid (PLA)

Afonso, J.A., J.L. Alves, G. Caldas, B.P. Gouveia, L. Santana, and J. Belinha, Influence of 3D printing process parameters on the mechanical properties and mass of PLA parts and predictive models

Gonzalez Alvarez, A., P.L. Evans, L. Dovgalski, and I. Goldsmith, Design, additive manufacture and clinical application of a patient-specific titanium implant to anatomically reconstruct a large chest wall defect.

Travieso-Rodriguez, J.A., R. Jerez-Mesa, J. Llumà, G. Gomez-Gras, and O. Casadesus, Comparative study of the flexural properties of ABS, PLA and a PLA–wood composite manufactured through fused filament fabrication.

 ANSWER: Several of the references listed above have been added to update the introduction.